# Transforming Growth Factor-Beta Orchestrates Tumour and Bystander Cells in B-Cell Non-Hodgkin Lymphoma

**DOI:** 10.3390/cancers14071772

**Published:** 2022-03-31

**Authors:** Matthew A. Timmins, Ingo Ringshausen

**Affiliations:** 1Wellcome Trust/MRC Cambridge Stem Cell Institute, University of Cambridge, Cambridge CB2 0AH, UK; mat74@cam.ac.uk; 2Department of Haematology, Addenbrooke’s Hospital, Cambridge University Hospital, Cambridge CB2 0AH, UK

**Keywords:** TGFB, B-cell lymphoma, microenvironment

## Abstract

**Simple Summary:**

Transforming growth factor-beta (TGFB) is a ubiquitously expressed cytokine involved in numerous functions in both normal and cancer cells. Here, we review, for the first time, the evidence of how B-cell lymphoma cells respond to TGFB and utilise TGFB to modulate surrounding cells in the microenvironment. We highlight recent data supporting the bi-directional signalling between B-cell lymphoma cells and their microenvironment. Targeting TGFB signalling in B-cell lymphoma may provide a future therapeutic strategy but further research is required to understand how lymphoma cells interact in different microenvironmental contexts.

**Abstract:**

Transforming growth factor-beta (TGFB) is a critical regulator of normal haematopoiesis. Dysregulation of the TGFB pathway is associated with numerous haematological malignancies including myelofibrosis, acute myeloid leukaemia, and lymphoid disorders. TGFB has classically been seen as a negative regulator of proliferation in haematopoiesis whilst stimulating differentiation and apoptosis, as required to maintain homeostasis. Tumours frequently develop intrinsic resistant mechanisms to homeostatic TGFB signalling to antagonise its tumour-suppressive functions. Furthermore, elevated levels of TGFB enhance pathogenesis through modulation of the immune system and tumour microenvironment. Here, we review recent advances in the understanding of TGFB signalling in B-cell malignancies with a focus on the tumour microenvironment. Malignant B-cells harbour subtype-specific alterations in TGFB signalling elements including downregulation of surface receptors, modulation of SMAD signalling proteins, as well as genetic and epigenetic aberrations. Microenvironmental TGFB generates a protumoural niche reprogramming stromal, natural killer (NK), and T-cells. Increasingly, evidence points to complex bi-directional cross-talk between cells of the microenvironment and malignant B-cells. A greater understanding of intercellular communication and the context-specific nature of TGFB signalling may provide further insight into disease pathogenesis and future therapeutic strategies.

## 1. Introduction

In recent years, the treatment landscape of B-cell non-Hodgkin lymphoma (B-NHL) has expanded to include targeted therapies antagonising the B-cell receptor pathway or antiapoptotic signalling. Despite impressive responses in some subtypes of B-cell lymphoma, a large fraction of patients inevitably relapses. The tumour microenvironment provides protective niches to malignant cells, shielding them from immune clearance and therapy [1]. Characterising the tumour microenvironment is imperative to understanding lymphomagenesis and drug resistance. Recent study of solid malignancies highlights transforming growth factor-beta (TGFB) as an influential factor in remodelling the tumour microenvironment, although less is known in B-cell malignancies [2,3]. Here, we review TGFB signalling and examine the role it plays in regulating haematopoiesis, normal B-cell development, and how B-NHL cells ultimately modulate this axis during tumourigenesis (Figure 1).

## 2. TGFB Isoforms, Secretion, and Activation

The TGFB superfamily of cytokines is an evolutionally conserved family of proteins found in all multicellular organisms. The superfamily is composed of a diverse number of ligands and includes TGFB, bone morphogenetic proteins (BMPs), inhibins, activins, and growth differentiation factors (GDFs). TGFB signalling is implicated in multiple cellular processes including growth, migration, extracellular matrix remodelling, invasion, epithelial–mesenchymal transition, and immune suppression. Although dynamically regulated under normal homeostasis, TGFB is frequently overexpressed in diseased states including malignant, inflammatory, and fibrotic pathologies.

Three isoforms—namely, TGFB1, TGFB2, and TGFB3—are synthesised as prohormones in mammalian cells. Their respective genes encode a large N-terminal portion termed the latency-associated peptide (LAP), a signal peptide, and a short C-terminal segment that corresponds to the mature active cytokine [4,5] (Figure 2). The signal peptide directs TGFBs to the endoplasmic reticulum where it is removed [4]. The remaining polypeptide translocates to the endoplasmic reticulum where they form a dimer that is subsequently cleaved by the endoprotease furin [6]. Following furin-mediated cleavage, the C-terminal fragment remains noncovalently associated with the disulphide linked homodimer of the LAP forming the small latent complex (SLC) [4]. In most cells, two copies of the N-terminal LAP are cross-linked with another small family of proteins termed latent TGFB binding proteins (LTBPs) with the combined complex termed the large latent complex (LLC) (Figure 2) [4]. This allows stabilisation via interaction with other components of the extracellular matrix (ECM). Overall, the three TGFB isoforms are highly homologous, with the active cytokine region sharing 71–79% amino acid sequence identity, whereas the LAP regions share 31–51% sequence homology [4]. Although similar effects of isoforms have been seen in vitro, distinct phenotypes have been observed in vivo. Germline TGFB1 KO mouse models are associated with inflammatory disease, defective haematopoiesis, and vasculogenesis, TGFB2 KO models are associated with perinatal mortality with multiple developmental abnormalities, and TGFB3 KO models are associated with cleft palate [7,8,9,10]. Accordingly, there is evidence that TGFB isoforms are differentially expressed in tissues [11]. Despite this, the majority of studies related to both normal haematopoiesis and haematological malignancies have focused on TGFB1, while the relative roles of TGFB2 and TGFB3 are less explored.

TGFB activation requires the release of LLC from the ECM, followed by proteolysis of LAP, leading to the release of active TGFB (Figure 2). Consequently, modulation of TGFB activity can be through direct changes in expression of TGFB or of proteins that interact directly or indirectly with TGFB. Mechanisms of activation of latent complexes include enzymatic matrix metalloproteinases (MMP) (MMP2 and MMP9), binding of thrombospondin (THBS1), and integrins [12,13,14,15,16]. Following activation, all isoforms interact with similar cell surface receptors and signal through intracellular signal cascades broadly defined as canonical and non-canonical pathways.

## 3. Canonical and Non-Canonical TGFB Signalling

TGFB ligands bind to the TGFB receptor complex composed of a tetrameric structure made up of two type-1 TGFB receptors (TGFBR1) and two type-2 TGFB receptors (TGFBR2). Less studied co-receptors—TGFBR3 (betaglycan) and endoglin—modulate the response of the receptor complex to TGFB ligands [17,18]. Engagement of the TGFB receptor complex leads to recruitment of intracellular receptor SMAD proteins (R-SMADs) to the cytoplasmic domain of TGFBR1, leading to their phosphorylation (Figure 3). Traditionally SMAD2 and SMAD3 are associated with TGFB, activin, and nodal signalling, whereas SMAD1, SMAD5, and SMAD8 are associated with BMP, GDF, and anti-Müllerian hormone signalling [19]. However, the binding of TGFB is able to also phosphorylate SMAD1 and SMAD5 in some cell types including epithelial, endothelial, fibroblast, and B-cell lymphoma cells [20,21,22,23]. Different type-1 receptors are traditionally associated with SMAD2/3 or SMAD1/5/8 phosphorylation, although there is evidence of interaction between type-1 receptors facilitating alternative phosphorylation cascades [23]. ACVR1B (ALK4), TGFBR1 (ALK5), and ACVR1C (ALK7) are generally associated with SMAD2/3 phosphorylation and ACVRL1 (ALK1), ACVR1 (ALK2), BMPR1A (ALK3), and BMPR1B (ALK6) associated with SMAD1, SMAD5, and SMAD8 phosphorylation [19,23]. Once R-SMADs are phosphorylated, a trimeric structure is formed with the common mediator SMAD4, facilitating translocation to the nucleus and regulation of gene expression through binding to SMAD-responsive regulatory regions. Collectively, this cascade is termed the canonical pathway; however, SMAD-independent signalling is recognised and is composed of alternative signalling pathways, including mitogen-activated protein kinases (MAPKs), phosphoinositide 3-kinase (PI3K), and Rho family of GTPases (Rho GTPase) (Figure 3) [19]. Targeting of some of these non-canonical pathways (e.g., PI3K) is established in the B-cell malignancy chronic lymphocytic leukaemia (CLL); however, the relevant contribution of autocrine or paracrine TGFB signalling in this cascade is unclear [24].

Following TGFB ligand-receptor binding several mechanisms can modulate the strength of downstream signalling including inhibitory SMADS (i-SMADs)—namely, SMAD6 and SMAD7. SMAD7 can bind to TGFBR1 and inhibit R-SMAD binding [25]. In addition, it can also promote TGFBR1 inactivation through the regulation of phosphorylation and recruitment of ubiquitin ligases [25,26,27,28]. Similarly, SMAD6 binds to TGFB type-1 receptors, where it inhibits SMAD1 and SMAD2 phosphorylation [29]. Finally, there are multiple non-i-SMAD inhibitory pathways, including modulation of phosphorylation of R-SMADS and regulation of TGFB receptors at transcriptional, translation, and post-translational levels [30,31]. As TGFB signalling is essential for tissue homeostasis, the pathway is tightly regulated at multiple levels and plays an important role in haematopoiesis and the tumour microenvironment.

## 4. TGFB and Haematopoiesis

Regulation of haematopoiesis relies on the balance of proliferation, differentiation, and apoptotic signals of haematopoietic progenitor cells. TGFB plays an important role but demonstrates cell and context-specific effects. Assays of CD34+ human stem cells treated with exogenous TGFB1 in vitro lead to growth arrest and inhibition of colony formation [32,33]. TGFB1 appears to preferentially maintain quiescence of early haemopoietic progenitors [32]. Notably, conditional knockout of TGFBR1 in adult mice leads to normal numbers of functional haemopoietic progenitor cells with normal self-renewal and regenerative capacities [34]. This may reflect TGFB1 signalling through alternative receptors, which has been reported in other cell types [35]. Alternatively, other cytokines or growth factors may compensate for the loss of TGFBR1. Whilst TGFB3 appears to be a potent negative regulator of haematopoietic stem cells (HSCs), TGFB2 displays more modest negative regulatory potential and, in some studies, even promotes HSC activity [36,37]. Clearly, there is a complex interplay between TGFB isoforms and other superfamily members to maintain haematopoiesis and the HSC pool. Understanding the impact of TGFB on normal haematopoiesis is important as relapse of CLL ultimately leads to bone marrow (BM) failure and death. Indeed, CLL skews the differentiation of HSCs towards myeloid progenitors at the expense of erythroid progenitors in the peripheral circulation [38]. Importantly, TGFB1 promotes myeloid-biased HSCs in vitro and in vivo [39]. Disease-induced alterations in the BM niche, mediated in part by TGFB, could therefore play a role in the observed redistribution of HSCs in CLL.

## 5. TGFB and the Bone Marrow Niche

The bone marrow niche is composed of a diverse set of cells that both secrete and modulate the TGFB milieu. Large quantities of latent TGFB are deposited in the bone matrix; however, relatively few BM cells exhibit significant phospho-SMAD2/3 (pSMAD2/3) expression in health suggesting tight spatial regulation [40]. Non-myelinating Schwann cells that lie parallel to blood vessels have been proposed to regulate TGFB activation through binding of latent TGFB via integrin B8, promoting activation through proteolytic cleavage by MMPs [40]. HSCs are frequently found in direct contact with these cells, and denervation leads to a rapid loss of HSCs from the BM, suggesting that glial cells maintain HSC dormancy through regulating activation of TGFB [40]. Indeed, TGFBR2 deficient HSCs have low-level SMAD activation, with reduced long-term repopulation capacity, highlighting the role of TGFB in HSC maintenance [40]. Megakaryocytes (MKs) have also been reported to regulate HSCs through the secretion of TGFB [41]. MK ablation leads to a reduction in active TGFB1 in the bone marrow, less pSMAD2/3 in HSCs and is associated with HSC activation and proliferation [41]. MK-derived TGFB1, therefore, appears to be an important signal in HSC quiescence.

Bone-marrow-derived stromal cells (BMSCs) are also important players in the BM niche. Beyond modulating haematopoiesis, they can play fundamental roles in lymphomagenesis and are capable of enhancing malignant B-cell survival, migration, as well as protecting against spontaneous and drug-induced apoptosis [42,43,44,45]. One of the hallmark functions of BMSCs is the ability to differentiate into osteoblasts, adipocytes, chondrocytes, and fibroblasts to maintain the BM microenvironment [46]. TGFB, in co-operation with other secreted factors, mediates these differentiation processes with the induction of fibroblast and chondrocyte differentiation and inhibition of adipocyte differentiation [47,48]. Some carcinomas appear to remodel their stromal microenvironment through overexpression of TGFB to form a desmoplastic stromal environment, inducing myofibroblast differentiation and ECM remodelling [49,50]. Co-culture of stromal fibroblasts and colon carcinoma cells induces the expression of numerous MMPs and TGFB target genes, leading to the formation of cancer-associated fibroblasts (CAF)-like cells [51]. Furthermore, CAFs have been shown to promote tumour progression in a TGFB-dependent manner [52]. Finally, TGFB modulates angiogenesis with TGFB deficient mice displaying defective endothelial differentiation [9]. Although less studied, the endothelial compartment and associated angiogenesis are influential in cancer progression and immunological response [53,54,55]. Although the majority of findings from solid malignancies show TGFB signalling to have a tumour-promoting role in stromal cells, other research indicates a tumour-suppressive role. Deletion of TGFBR2 in mouse fibroblasts leads to epithelial neoplasia, with subsequent studies showing that loss of TGFB signalling increases the progression of several solid cancers [56,57,58]. Overall, TGFB signalling is critical for various fibroblast functions, including secretion of chemokines, growth factors, matrix deposition, and ECM remodelling; nevertheless, it appears to be tightly regulated dependent on cellular context.

The effect of TGFB signalling on modulation of chemokine and cytokine secretion by BMSCs highlights a role for TGFB in the immune response. Indeed, lessons from murine models and solid malignancy highlight TGFB as a regulator of activation, recruitment, and function of multiple cells of the innate and adaptive immune system. Generally, TGFB has an immunosuppressive effect on the innate system. TGFB signalling supports M2 macrophage polarisation associated with the promotion of angiogenesis and tumourigenesis [59,60]. Similarly, in neutrophils, TGFB leads to immunosuppressive N2 neutrophil expansion and inhibition of natural killer cells (NK) maturation, impairing recognition and clearance of malignant cells [61]. TGFB1 also plays a key regulatory role in the adaptive immune system with a multiorgan autoimmune phenotype observed when TGFB1 signalling is removed from T-cells [62]. Dependent on the presence of secreted factors and expression of cell surface co-receptors, TGFB can suppress the adaptive immune system by promoting T-regulatory and suppressing Th1, Th2, and CD8+ cytotoxic cells or enhance their response through induction of Th17, Th9, and CD4 cytotoxic-like effector cells [4,63]. Ultimately, TGFB, in concert with other factors, orchestrates dynamic immune responses through complex cross-talk with other cells of the BM niche in a process termed osteoimmunology [64,65]. Indeed, there is a growing appreciation of the cellular diversity of the BM niche made possible through advances in single-cell technologies with 17 distinct subsets of mouse BM stroma cells alone [66]. It is, therefore, helpful and necessary to dissect TGFB signalling in a cell-context-specific manner in their respective relevant microenvironment.

## 6. TGFB and Normal B-Cells

Accumulating evidence highlights TGFB as an important regulator of B-cells including development, activation, and differentiation. Some of these effects are direct actions of TGFB but some aspects of B-cell response are indirect through intercellular communication.

Beyond regulating the HSC pool, TGFB modulates subsequent B-cell development and activation. TGFB1 inhibits kappa-light-chain expression in murine pre-B-cell clones exerting an inhibitory selective effect on the acquisition of surface gamma chains [67,68]. Furthermore, TGFB has been linked to the control of the germinal centre (GC) [69,70]. Mice lacking TGFBR2 specifically in GC B-cells, display hyperproliferation of centroblasts, centrocytes, and GC cells [70]. Nevertheless, different murine models have shown contrasting results of TGFB in B-cell responsiveness and immunoglobulin production [71,72,73].

TGFB clearly plays a role in regulating B-cell proliferation and survival. In vitro TGFB1 triggers apoptosis in normal B-cells and some transformed B-cell lines, leading to induction of proapoptotic and downregulation of antiapoptotic proteins [74,75,76,77]. However, whilst in vivo mouse models of B-cell selective TGFBR2 deficiency show decreased life span of conventional B-cells, an accumulation of peritoneal B1 cells, as well as B-cell hyperplasia in Peyer’s patches has been observed in the same model [78]. This suggests that TGFB signalling may have both pro- and antiapoptotic effects in B-cells, depending on the context, stage of development, and microenvironment. Collectively, these studies highlight the complexity of TGFB B-cell biology and reflect the relatively poorly understood interactions between TGFB signalling and normal B-cell regulation [79]. Despite this, accumulating evidence points towards TGFB dysregulation in the B-cell malignancies.

## 7. Intrinsic Dysregulation in Malignant B-Cells

Most studies of TGFB dysfunction in haematological malignancies have focussed on myeloid malignancies, given a higher frequency of specific oncoproteins and mutations in TGFB signalling components (reviewed by Bataller et al. [80]). Nevertheless, TGFB dysregulation has been found in a number of B-cell malignancies. TGFB and related signalling components are overexpressed in CLL, follicular lymphoma (FL), hairy cell leukaemia (HCL), and mantle cell lymphoma (MCL), as well as multiple myeloma (MM) [81,82,83,84,85,86,87,88,89,90]. Dysregulation occurs at multiple levels within the signalling cascade, including receptor expression, SMAD signalling, epigenetic and genetic mechanisms (Figure 4).

Resistance to apoptotic and antiproliferative effects of TGFB in CLL has been attributed to the loss of surface TGFB receptors [85,91,92]. Notably, not all studies find decreased surface expression of TGFB receptors, with some studies reporting elevated TGFBR3 surface expression yet the significance of this is unclear [93,94]. Intriguingly, Epstein–Barr virus (EBV) status plays an important role in TGFBR expression, with EBV transformed B-cell lines resistant to antiproliferative effects of TGFB1 through downregulation of TGFBR1 or TGFBR2 [95,96,97]. TGFBR2 expression may play a broader role in B-cell malignant development. TGFBR2 is a component of gene expression signatures of diffuse large B-cell lymphoma (DLBC) subclasses, with GC-like DLBCL showing a relative decrease in TGFBR2 expression, whereas activated B-cell (ABC)-DLBCL shows relative increase [98]. Consistent with this, TGFBR2 expression forms part of the DLBCL consensus gene signature with higher expression in the host response cluster of DLBCL, compared with the oxidative phosphorylation cluster and B-cell receptor/proliferation cluster [99]. Collectively, this points to TGFB dysfunction playing a potential role in the pathogenesis of distinct molecular subtypes of DLBCL.

Although the literature is limited, genetic and epigenetic changes have been identified related to TGFB-receptor regulation. Mutations within the putative signal sequence of the TGFBR1 gene have been identified in a small cohort of CLL patients and found to correlate with resistance to growth inhibitory effects of TGFB [100]. Similarly, downregulation of TGFBR2 via promoter methylation in B-cell lymphomas confers resistance to TGFB mediated growth suppression, which is partially reversible in vitro with the demethylating agent azacitidine [101]. Intriguingly, treatment of a resistant B-cell lymphoma cell line with anti-IgM upregulated TGFBRs and restored growth inhibitory effects of TGFB but not TGFB apoptosis, suggesting an independent mechanism of apoptotic regulation [102]. Mechanistic study of TGFB mediated apoptotic signalling in malignant B-cells is lacking and represents an area in need of further research particularly given advances in targeted therapies of the apoptotic axis such as BCL2 inhibitors (venetoclax). Small studies predominantly limited to Burkitt lymphoma (BL) cell lines suggest TGFB may regulate the expression of apoptotic regulators BIK and Bcl-xL [75,77]. Interestingly, gene expression profiling of primary leukemic phase MCL cells highlights the aberrant expression of genes associated with TGFB signalling pathway activation, including downstream targets associated with resistance to FAS-mediated apoptosis (e.g., CASP8 and FADD-like apoptosis regulator (CFAR)) [81,103]. Thus, it seems likely that TGFB regulates malignant B-cell apoptotic response dependent on malignant cell type and context.

Although earlier research focussed on receptor-mediated resistant mechanisms, a more recent study has revealed distinct intracellular signalling alterations in follicular lymphoma and DLBCL. SMAD1 appears increasingly important in mediating TGFB signalling in malignant B-cells. Gene expression profiling highlights SMAD1 as overexpressed in FL and EBV negative BL cells [83,104]. Indeed, SMAD1 is the most differentially overexpressed gene in FL cells, compared with normal GC B-cells [83]. Although typically associated with BMP signalling (Figure 3), Munoz et al. described TGFB mediated SMAD1 signalling in FL cells which previously had only been reported in endothelial cells and epithelial cells [21]. Moreover, SMAD1 modulates lymphoma cell’s response to TGFB with SMAD1 knockdown or mutational inactivation associated with resistance to antiproliferative effects of TGFB [21]. What triggers SMAD1 induction, and through which receptors TGFB1-SMAD1 signals remains to be experimentally determined. Nevertheless, as SMAD1 has distinct transcriptional targets, compared with other R-SMADs in other cell types, SMAD1 likely mediates a transcriptional programme specific to FL B-cells linked to pathogenesis [23,105].

In contrast to FL, aberrant loss of SMAD1 is common in DLBCL and found in >85% of DLBCL patients, likely reflecting differing fundamental biology [70]. Mechanistically, TGFB signalling via the TGFBR2–SMAD1 axis regulates expression of the tumour suppressor sphingosine-1-phosphate receptor (S1PR2), controlling DLBCL survival in vitro and restricting tumour growth in vivo [70]. Chemoresistance in DLBCL is associated with aberrant DNA methylation of SMAD1. Reactivation of SMAD1 with demethylating agents restores SMAD1 expression and is associated with chemosensitisation in vitro and in vivo [70,106,107].

Beyond perturbed SMAD1 signalling, DLBCL cells can also mediate TGFB signalling through SMAD5 phosphorylation, which is again more commonly associated with BMP signalling [108]. This surprising finding by Rai et al. emerged as a result of a study of microRNA-155 in DLBCL [108]. This small regulatory noncoding RNA is overexpressed in a subset of DLBCL patients and associated with aggressive disease [109,110]. MicroRNA-155 binds to the SMAD5 gene and suppresses expression, leading to resistance of cytostatic effects of TGFB and more aggressive disease in xenograft models [108]. Collectively, these studies highlight the complexity of intracellular TGFB signalling in B-cell malignancies with multiple heterogeneous mechanisms of perturbed signalling dependent on histological subtype. Studies predominantly reveal tumour intrinsic resistant mechanisms to growth inhibitory or proapoptotic effects of TGFB1. These mechanisms may confer a selective advantage for clonal expansion through direct, but also indirect, effects on other cells in the microenvironment.

## 8. Microenvironmental TGFB in B-Cell Malignancies

The B-cell malignant microenvironment is made up of multiple different cell types whereby complex reciprocal signalling can occur through both tumoural and microenvironmental TGFB (Figure 4). We focus on T-cells, stromal cells, and innate immune cells for which there is direct evidence of the role of TGFB in B-cell malignancies. However, it is likely other cells of the microenvironment are impacted and potentially modulate TGFB. Indeed, endothelial cells are important players in the malignant niche; however, the role TGFB plays in modulating these cells in B-cell lymphomas remains to be experimentally determined [111,112,113].

### 8.1. T-Cells

The role of microenvironmental TGFB has been explored most in FL and has been linked to both stromal and T-cell reprogramming. Both FL cells and intratumoural T-cells secrete TGFB, which promotes regulatory CD4+ T-cell expansion and suppresses effector helper CD4+ cell function [114]. In addition to upregulating FoxP3 and IL2, TGFB induces activation of CD70 preferentially in intratumoural T-memory cells mediating functional exhaustion [115]. Accordingly, the frequency of CD70+ T-cells is associated with poorer survival in FL [115]. Whilst soluble TGFB has been the focus of most investigations related to tumour mediated T-cell polarisation, there is an increasing appreciation that cells can harbour functional membrane-bound TGFB [116,117]. Interestingly B-cell lymphoma cells, but not normal B-cells or other cells of the microenvironment, express membrane-bound TGFB capable of suppressing intratumoural T-cells in vitro [114]. Notably, this surface binding is mediated by non-TGBR mechanisms involving heparan sulphate proteoglycans [114]. Indeed, similar findings have been found in cutaneous T-cell lymphoma [118]. Thus, both soluble and membrane-bound TGFB lead to immunosuppressive T-cell polarisation in FL. Importantly, T-cell dysfunction is widely recognised in multiple other TGFB secreting B-cell malignancies such as CLL and DLBCL, but there is a paucity of data in exploring TGFB in these contexts.

### 8.2. Stromal Cells

More recent reports highlight complex cross-talk between malignant B-cells and stromal cells. FL B-cells produce extracellular vesicles (EVs) that are internalised by BMSCs, drive stromal polarisation, and enhance FL B-cell survival and quiescence [82]. Examining signalling alterations triggered by these EVs highlights a dominant role of TGFB signalling. EVs from FL cells contain higher levels of TGFB1, compared with tonsil B-cell-derived EVs, and activate canonical and non-canonical signalling in stromal cells [82]. As EVs are shed into circulation and provide a method of cell contact independent stromal polarisation, it is reasonable to hypothesise that malignant B-cell-derived EVs may prime distant non-invaded niches for subsequent invasion. A more recent study has highlighted bidirectional TGFB signalling between FL cells and lymphoid stromal cells [119]. Stroma-mediated TGFB signalling, along with other chemokines, are predicted to drive a distinct FL B-cell transcriptional programme [119]. Accordingly, SMAD1, SMAD3, and SMAD4 are upregulated in FL cells, compared with normal centrocytes [119]. Conversely, TGFB and TNF signalling pathways are the most discriminatory drivers of FL lymphoid stromal polarisation [119]. Importantly, TGFB and TNF signalling synergise in driving LSC polarisation of putative lymphoid stroma precursors in vitro, reflecting the importance of how other secreted factors coordinate TGFB cellular response. Malignant B-cells are known to remodel stromal cells in other low-grade B-cell malignancies such as CLL to form protective niches. In CLL, this is partly mediated through the induction of stromal protein kinase C beta (PKC-β) [111]. Although not functionally characterised, multiple TGFB mediated signalling components are predicted to regulate stromal PKC-β-dependent drug resistance in CLL cells [120]. Consistent with the role of TGFB remodelling the CLL niche is the observation that TGFB1 is secreted at higher levels from CLL BM stromal cells, compared with normal BM stromal cells [87]. Furthermore, bone marrow plasma from HCL patients induces fibrosis of normal BM stroma in vitro and is neutralised by TGFB1 antibodies, highlighting the role of TGFB in the modulation of the stromal microenvironment of lymphoproliferative disorders [84].

The role TGFB plays in the microenvironment of DLBCL is less clear. Early sequencing studies identified enrichment of TGFB1 signalling elements within prognostically favourable ‘stromal 1’ signature [113]. A more recent large sequencing study of over 4000 DLBCLs has further dissected the microenvironment into four microenvironmental signatures that predict prognosis [121]. TGFB1 pathway activity was elevated in the ‘mesenchymal (MS) lymphoma microenvironment’ and reduced in the ‘depleted (DP) lymphoma microenvironment’, highlighting how TGFB may modulate local cellular composition [121]. The MS lymphoma microenvironment was associated with better overall survival, compared with the DP microenvironment [121]. Accordingly, DP DLBCL was enriched for SMAD1 methylation, providing a mechanism for DLBCL to overcome the potential anti-lymphoma role of TGFB mediated secreted by surrounding stroma [121]. Thus, TGFB may mediate pro- and anti-lymphoma effects based on DLBCL cell-intrinsic and -extrinsic factors, underlining the need to study on a cell and microenvironmental context.

### 8.3. Innate Immune Cells

Finally, microenvironmental TGFB orchestrates aspects of innate immune function. B-cell acute lymphoblastic leukaemia (ALL)-derived TGFB1 mediates natural killer cell dysfunction, demonstrating impaired interferon-gamma secretion and cytotoxicity, allowing ALL to evade immune clearance [122]. Conversely, other innate cells reprogramme malignant B-cells via TGFB to limit tumourigenesis. A study of a transgenic mouse model of Myc-driven lymphomagenesis demonstrates apoptotic lymphoma cells activate macrophages, leading them to secrete TGFB1, which drives lymphoma cell senescence [123]. Thus, macrophage-derived TGFB signals may limit malignant progression through senescence induction. Importantly, although malignant cell senescence is generally considered favourable, it should be considered that the long-term survival of senescent cells may seed relapse and more aggressive disease [124].

Overall microenvironmental TGFB predominantly leads to immune cell suppression and stromal reprogramming, leading to a protumoural niche. Moreover, there is increasing evidence of bi-directional cross-talk, whereby cells of the microenvironment promote senescence and survival in malignant B-cells. Although these aberrations should be considered on a disease-subtype-specific basis, it is likely that some of these changes promote tumour growth and seed relapse, offering a potential new route to therapy.

## 9. Therapeutic Perspective

A challenge to therapeutic translation of strategies targeting the TGFB axis is the heterogeneous mechanisms of dysregulation. One approach would be to reactivate sensitivity to the proapoptotic effects of TGFB. The epigenomic evolution of DLBCL is associated with the methylation of TGFB signalling elements, offering the opportunity to utilise demethylating agents. Notably, pre- and post-treatment biopsies from a small cohort of patients treated with azacytidine, followed by standard chemoimmunotherapy in high-risk, newly diagnosed DLBCL, confirm SMAD1 demethylation and associated chemosensitisation [107]. Nevertheless, the relative contribution of other aberrantly methylated genes in other pathways in this process is unclear. Furthermore, the heterogeneous nature of DLBCL provides challenges to this approach, with distinct DLBCL microenvironmental signatures associated with different TGFB1 signalling activities.

An alternative approach to modifying intrinsic TGFB dysregulation is to inhibit prosurvival microenvironment signalling. Targeting these interactions is advantageous as microenvironmental cells are not under selective pressure. For example, microenvironmental disruption of the CLL niche potentiates the antitumour activity of Ibrutinib, and targeting stromal PKC-β in B-cell lymphomas enhances toxicity to common therapies [120,125]. Targeting TGFB signalling between FL B-cells and stroma may perturb the establishment of favourable stromal niches but also reverse immunosuppressive T-cell response to synergise with therapy. Coherent with this rationale are findings from solid malignancies. Single-cell sequencing study of solid malignancies finds stromal cells with enriched TGFB signalling associated with primary resistance to immunotherapy [2,126]. TGFB blocking antibodies synergise with immunotherapy, reduce TGFB signalling in stromal cells, and enhance T-cell penetration of tumours in mouse models [127]. Targeting the TGFB axis has advanced rapidly in solid malignancies with phase II trials of a small molecule inhibitor of TGFBR1, demonstrating improved overall survival in hepatocellular carcinoma and unresectable pancreatic cancer [128,129]. However, targeting the TGFB axis presents many challenges that are shared with solid cancers including the dynamic and dual roles of TGFB signalling (reviewed by Teixeira et al. (2020) [130]). An interesting approach to mitigate broader effects of TGFB inhibition could be to target local regulators of latent TGFB activation. In multiple myeloma (MM), thrombospondin1 (THBS1) is overexpressed and activates latent TGFB [131] (Figure 2). Targeting THBS1 with an antagonistic peptide in MM mouse models synergises with cytotoxic therapies [131]. Such approaches targeting tumour-derived regulators of TGFB activation may mitigate the effects of broader TGFB inhibition strategies. Nevertheless, how TGFB activation may be perturbed in other B-cell lymphomas remains to be investigated.

Finally, the TGFB axis should be considered in the context of the expanding number of cellular-based therapies for B-cell lymphomas. Early research using EBV-specific cytotoxic T-cell lymphocytes used for the treatment of EBV-positive Hodgkin disease show that modification through the expression of a dominant-negative TGFBR2 resisted the immunosuppressive effects of TGFB and enhanced cytotoxicity [132,133]. A similar approach is being tested with dominant-negative TGFBR2 CAR-T cells in prostate cancer [134]. Such approaches or combination immunotherapy may be necessary for CAR-T- and NK-cellular-based therapy against advanced immunosuppressive tumours. Nevertheless, given potential pro- and anti-lymphoma effects of TGFB, detailed, functional characterisation of malignant B-cells and surrounding microenvironment is required to therapeutically target the TGFB axis. Ultimately, future clinical trials need to incorporate prospective characterisation of both malignant B-cells and cells of the tumour microenvironment. Whilst this has traditionally been challenging, advances in single-cell technologies make this increasingly feasible.

## 10. Conclusions

Multiple mechanisms of intrinsic TGFB signalling dysfunction have been identified in B-cell malignancies with distinct alterations dependent on histological subtype. Resistance to tumour-suppressive mechanisms of TGFB may confer a selective advantage for clonal expansion through direct, but also indirect, effects on other cells in the microenvironment. Inhibiting reciprocal TGFB signalling between B-cells and microenvironmental cells offers a potential therapeutic window to target protumoural niches. Given the cell-context nature of TGFB signalling, future meticulous functional characterisation of the B-cell tumour microenvironment may provide insights into disease pathogenesis and guide therapeutic strategies.

## Figures and Tables

**Figure 1 cancers-14-01772-f001:**
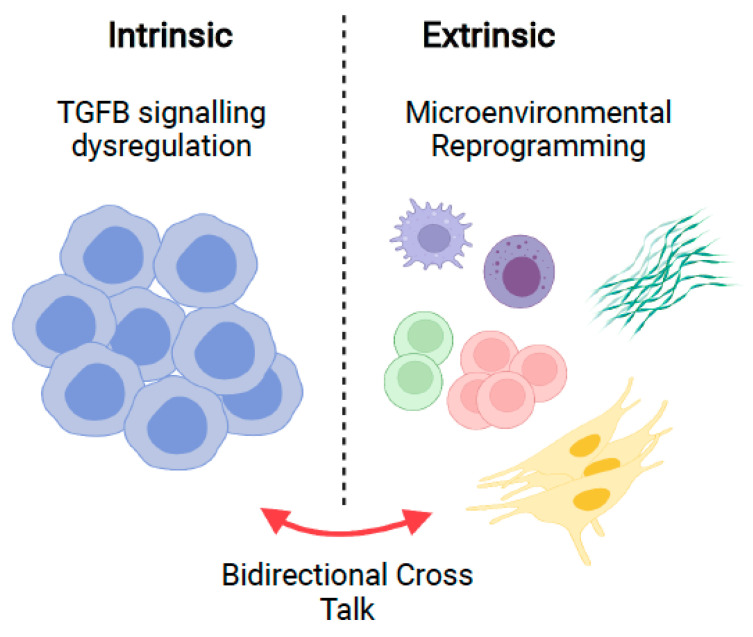
Bone marrow niche effects of TGFB. Heterogenetic regulation of tumour intrinsic signalling pathways and reprogramming of extrinsic bystander cells by TGFB promote disease progression of B-cell malignancies. Figure generated using Biorender.com (accessed 28 February 2022).

**Figure 2 cancers-14-01772-f002:**
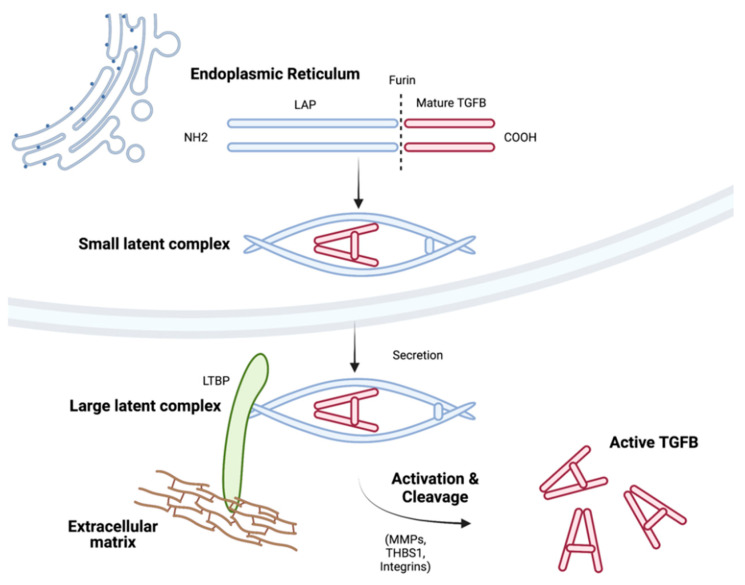
Processing of TGFB isoforms. TGFB genes encode a C-terminal mature cytokine and an N-terminal latency-associated protein (LAP) that form a dimeric structure. LAP and mature TGFB cytokine are cleaved by the enzyme furin but remain non-covalently associated with LAP folding around the mature cytokine forming the small latent complex. The small latent complex can associate with latent TGFB binding protein (LTBP) via LAP forming the large latent complex facilitating binding to the extracellular matrix. For active signalling TGFB must be released from latent complexes mediated through interactions with MMPs, THBS1, and integrins [4]. Figure generated using Biorender.com (accessed 28 February 2022).

**Figure 3 cancers-14-01772-f003:**
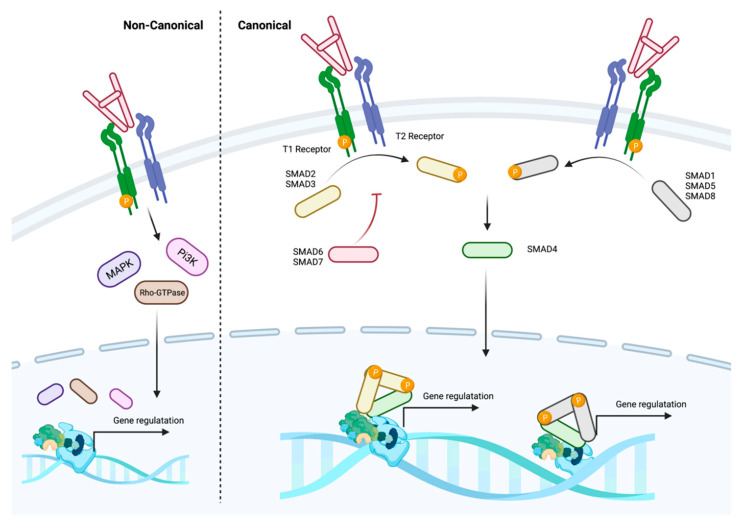
Overview of core TGFB signalling. In the canonical pathway, active TGFB binds to preformed dimers of type-1 (TGFBR1) and type-2 (TGFBR2) receptors to form a heterotetrametric active receptor complex. The constitutively active type-2 receptor activates type-1 receptor kinase activity, allowing recruitment and phosphorylation of R-SMADs and the formation of complexes with SMAD4. These SMAD4 complexes translocate to the nucleus and regulate gene expression in combination with transcription factors, histone-modifying enzymes, and chromatin remodelling complexes. TGFB receptor activation can also trigger non-SMAD-mediated signalling (non-canonical) including activation of MAPK, PI3K, or Rho GTPases. Figure generated using Biorender.com (accessed 28 February 2022).

**Figure 4 cancers-14-01772-f004:**
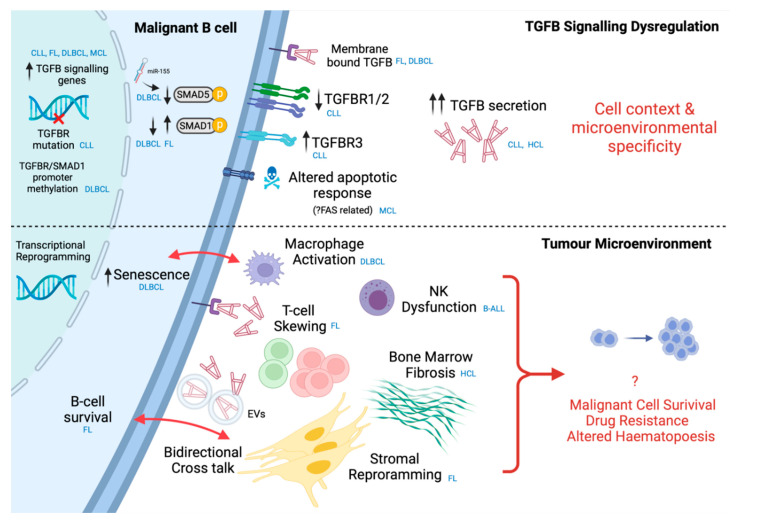
Overview of intrinsic and extrinsic TGFB signalling alterations in B-cell malignancies. Upper panel: Intrinsic signalling alterations in B-cell malignancies ranging from receptor dysfunction, SMAD signalling alterations, and genetic and epigenetic alterations. Distinct alterations are observed depending on the histological subtype defined in blue and are likely dependent on the microenvironmental context. Lower panel: Extrinsic effects of microenvironmental TGFB on B-cell tumour microenvironment. TGFB predominantly promotes a prosurvival niche mediated by immune suppression and stromal reprogramming. Complex bidirectional cross-talk between cells of the microenvironment (stroma and macrophages) drives senescence and survival in malignant B-cells. Alterations observed are dependent on the histological subtype defined in blue. Overall, intrinsic resistance to TGFB and TGFB mediated protumour microenvironmental reprogramming may drive malignant cell survival, drug resistance, and disturbed haematopoiesis. CLL: chronic lymphocytic leukaemia, DLBCL: diffuse large B-cell lymphoma, FL: follicular lymphoma, MCL: mantel cell lymphoma, B-ALL: B-cell acute lymphoblastic lymphoma. Figure generated using Biorender.com (accessed 28 February 2022).

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
