# Peer review of "Transforming Growth Factor-Beta Orchestrates Tumour and Bystander Cells in B-Cell Non-Hodgkin Lymphoma"

_cancers, 2022, doi:10.3390/cancers14071772_

Round 1

Reviewer 1 Report

The authors have clarified several of the questions I raised in my previous review. Most of the major problems have been addressed by this revision.

Reviewer 2 Report

This is a revised version and a significantly improved one.

This manuscript is a resubmission of an earlier submission. The following is a list of the peer review reports and author responses from that submission.

Round 1

Reviewer 1 Report

In this review the authors provide a comprehensive account of the role of targeting TGFB signalling in B-cell lymphoma. The following have not been addressed:

1) The rationale of why the authors came up with this review.

2) What is the information that is not exactly available that motivated the authors to come up with this information. What are the current caveats and how do the authors highlight the current research in answering them? If not they need to address in future directions.

3) In Section 5 "TGFB and The Bone Marrow Niche. The authors do not have any evidence for solid tumors to make the review global. With the current set of info, it infers only to some aspects of the angiogenetic process (reff 54-55). A broader standpoint should also consider the implications of the angiogenesis for a more modern applications: tumors grow and evolve through a constant crosstalk with the surrounding microenvironment, and emerging evidence indicates that angiogenesis and immunosuppression frequently occur simultaneously in response to this crosstalk. Accordingly, strategies combining anti-angiogenic therapy and immunotherapy seem to have the potential to tip the balance of the tumor microenvironment and improve treatment response (please refer to PMID: 33203154 and expand)

4) Does endothelial cells in angiogenesis in a tumor micro-environment involve hypoxia and TGFB pathway? Since hypoxia is a key factor for angiogenesis, the authors need to substantiate.

5) referring to point 4, especially in  DLBCL, angiogenesis, cell adhesion mediated drug resistance, tumor microenvironment, tumor progression play a pivotal role in shaping the microenvironment (i.e. refer to PMID: 32664527 or PMID: 30745366).

6) The authors need to highlight what new information the review is providing to enhance the research in progress.

Reviewer 2 Report

In their review Matthew A Timmins and Ingo Ringshausen provide key messages regarding Transforming growth factor beta (TGFB) is a critical regulator of normal hematopoiesis with a peculiar regards to NHLs.

Point to be addressed:

  1. The underlying message here is that more precision and individualized approaches need to be tested in well designed clinical trials – a challenge, but I would be interested in their perspective of how this might be done.
  2. As far as I understand the images (high quality!) are performed by using Biorender: this should be acknowledged, all users must cite BioRender figures with the credit “Created with BioRender.com.” 
  3. An explanatory graphical abstract summarizing the main findings and author flow across the review would better dive the reader into the 10 paragraphs.
  4. In section 5, TGFB in the Bone Marrow Niche this reviewer personally misses some insights regarding osteoimmunology, a term coined about twenty years ago to identify a strict cross talk between bone niche and immune system both in physiological and pathological activities, including hematological cancer. Several molecules are involved in the complex interaction between bone niche, immune and cancer cells. TGFB makes no exception. Can the authors expand referring to PMID: 32064051? This would increase the interest of this well written manuscript for a broad readership from the oncology field.